# Application of the NSGA-II Algorithm and Kriging Model to Optimise the Process Parameters for the Improvement of the Quality of Fresnel Lenses

**DOI:** 10.3390/polym15163403

**Published:** 2023-08-14

**Authors:** Hanjui Chang, Yue Sun, Rui Wang, Shuzhou Lu

**Affiliations:** 1Department of Mechanical Engineering, College of Engineering, Shantou University, Shantou 515063, China; 22ysun@stu.edu.cn (Y.S.); m18403985082@163.com (R.W.); 21szlu@stu.edu.cn (S.L.); 2Intelligent Manufacturing Key Laboratory of Ministry of Education, Shantou University, Shantou 515063, China

**Keywords:** Fresnel lens, residual stress, transmittance, NSGA-II algorithm, Kriging model, non-destructive analysis

## Abstract

The Fresnel lens is an optical system consisting of a series of concentric diamond grooves. One surface of the lens is smooth, while the other is engraved with concentric circles of increasing size. Optical interference, diffraction, and sensitivity to the angle of incidence are used to design the microstructure on the lens surface. The imaging of the optical surface depends on its curvature. By reducing the thickness of the lens, light can still be focused at the same focal point as with a thicker lens. Previously, lenses, including Fresnel lenses, were made of glass due to material limitations. However, the traditional grinding and polishing methods for making Fresnel lenses were not only time-consuming, but also labour-intensive. As a result, costs were high. Later, a thermal pressing process using metal moulds was invented. However, the high surface tension of glass caused some detailed parts to be deformed during the pressing process, resulting in unsatisfactory Fresnel lens performance. In addition, the complex manufacturing process and unstable processing accuracy hindered mass production. This resulted in high prices and limited applications for Fresnel lenses. These factors prevented the widespread use of early Fresnel lenses. In contrast, polymer materials offer advantages, such as low density, light weight, high strength-to-weight ratios, and corrosion resistance. They are also cost effective and available in a wide range of grades. Polymer materials have gradually replaced optical glass and other materials in the manufacture of micro-optical lenses and other miniaturised devices. Therefore, this study focuses on investigating the manufacturing parameters of Fresnel lenses in the injection moulding process. We compare the quality of products obtained by two-stage injection moulding, injection compression moulding, and IMD (in-mould decoration) techniques. The results show that the optimal method is IMD, which reduces the nodal displacement on the Fresnel lens surface and improves the transmission performance. To achieve this, we first establish a Kriging model to correlate the process parameters with optimisation objectives, mapping the design parameters and optimisation objectives. Based on the Kriging model, we integrate the NSGA-II algorithm with the predictive model to obtain the Pareto optimal solutions. By analysing the Pareto frontier, we identify the best process parameters. Finally, it is determined that the average nodal displacement on the Fresnel surface is 0.393 mm, at a holding pressure of 320.35 MPa and a melt temperature of 251.40 °C. Combined with IMD technology, product testing shows a transmittance of 95.43% and an optimisation rate of 59.64%.

## 1. Introduction

With the development of micro-optics, the applications of micro-lenses in biomedical and energy fields are increasing, among which Fresnel lenses are typical representatives of micro-lenses. The principle of a Fresnel lens is based on the Fresnel waveguide, which is similar to a lens that focuses incident light to produce a high intensity of light. Fresnel lenses, also known as threaded lenses, have a short focal length, require less material, and are lightweight. Compared to conventional lens designs, Fresnel lenses are thinner and can therefore transmit more light. The light is refracted by the Fresnel lens and can be focused more effectively at a distance. Fresnel lenses are used in a wide range of applications, including lighthouse navigation lights, projector lenses, rear projection displays, and automotive headlights. Most optical lenses are manufactured using injection moulding in conjunction with moulds, and conventional Fresnel lenses are manufactured mainly using this method. The moulding process for producing Fresnel lenses is not fundamentally different from the conventional mass production process and has little impact on the design of various lenses. However, the precision and quality of Fresnel lenses still depend on the processing method and injection moulding parameters. Therefore, by optimising the processing techniques and injection moulding parameters, the challenges of manufacturing non-traditional Fresnel lens functional surfaces can be overcome and their quality performance ensured.

In recent decades, Fresnel lens manufacturing technology has evolved to meet the demand for increasingly precise, complex, and miniaturised Fresnel lenses. This technological progress has driven the development of optical design, manufacturing, and testing of new Fresnel lenses. However, there are many new challenges to be faced and overcome in the pursuit of technological advances before the full potential of these lenses can be realised, which also brings new perspectives to the design, fabrication, and testing of lenses. Therefore, the purpose of this paper is to address the challenges and obstacles faced in the manufacture of Fresnel lenses.

Coating on optical polymers can improve the mechanical and optical properties of lenses. To improve light transmission and image quality, lenses are usually coated in modern lens manufacturing processes. According to the principle of optical interference, the reflection of light of a certain wavelength from the lens can be minimised by coating the lens with a thin film, which is a quarter of the thickness of the wavelength.

In-mould decoration (IMD) is an advanced manufacturing technology commonly used for surface decoration of plastic products. It combines injection moulding and pattern decoration, allowing patterns, colours, or other decorative elements to be permanently applied to the surface of plastic parts in a single process step.

When using the IMD process, the lack of thin film material directly affects the optical performance of the lens; therefore, the necessity of thin film quality cannot be ignored. Excellent film quality is not just a technical detail, it is a key factor in ensuring optimum lens functionality. During the injection moulding process, the quality and consistency of the lens part depend mainly on the precise control of the process conditions. This includes, but is not limited to, the precise setting of process parameters, such as temperature, pressure, and cooling rate, to ensure that the film covers the lens surface uniformly and without bubbles, wrinkles, or other defects. Additionally, the design of the mould is critical, taking into account the characteristics of the film material to ensure that the film is evenly distributed and well adhered to during the injection moulding process. By making these improvements, we can ensure that the optimum optical performance of the lens is achieved to meet demanding application scenarios and user expectations.

In this article, we use non-destructive testing and evaluation methods. The injection moulding process involves injecting the molten state of polymer resin mixed with additives into the mould cavity, allowing the resin to solidify, and then opening the mould to release the plastic component. While injection moulding itself has a low carbon footprint, we need to consider the impact of auxiliary processes and material consumption, as the production of raw materials generates carbon emissions. As injection moulding is a large-scale industry, these carbon emissions can have a significant environmental impact. By improving energy efficiency, we can effectively reduce such emissions.

By optimising the injection moulding process, we can achieve the best injection parameters to produce a large number of Fresnel lenses, thereby avoiding the waste resulting from poorly performing products, improving production efficiency and reducing carbon emissions. With changing demands for plastic products and advances in the development of biodegradable materials, injection moulding technologies in the industry continue to evolve towards high precision, high efficiency, low energy consumption, environmental safety, and intelligent automation.

As shown in Figure 1, a Fresnel lens is a device that converts a spherical or aspherical lens into a light and thin planar shape to achieve the same optical effect. The main difference between a conventional lens and a Fresnel lens is that a Fresnel lens achieves significant material savings, while maintaining the same light gathering effect, by removing the part of the conventional lens that propagates in a straight line and retaining only the curved surface where refraction occurs. Ultra-precision machining techniques are used to create a series of optical quality rings on the flat surface, each with an independent lens function. This makes Fresnel lenses the best choice for large, flat, and thin lenses. In short, a Fresnel lens is flat on one side and convex on the other. Because of this special structure, the manufacturing process of a Fresnel lens is more complicated and demanding than the traditional lens process. It requires excellent surface quality and the lowest possible product node displacement and residual stress to improve product performance and light transmission. Therefore, this paper proposes to optimise the injection moulding process of Fresnel lenses.

However, the existing injection moulding process for Fresnel lenses is not sufficiently mature. Incorrectly set injection parameters can lead to an uneven flow rate of the thermoplastic in the mould cavity, thus prolonging the moulding cycle and making the surface of the Fresnel lens prone to defective flow marks and air lines, which eventually leads to the failure to achieve a fully high-gloss mirror effect. In addition, due to the uneven processing temperature in different parts of the mould, the plastic material with the lower temperature cools down the mould first, resulting in short shot defects with insufficient filling in the sharp parts of the Fresnel lens. This Fresnel lens with short shot defects may also be accompanied by stress concentration issues, which further affects the optical imaging and seriously affects the product quality. In this paper, a multi-objective optimisation method based on the Kriging model and non-dominated genetic algorithm (NSGA-II) is proposed.

## 2. Literature Review

In 2012, Tosello et al. [1] coated tin on nickel inserts of polycarbonate micro-Fresnel lenses. The surface wear was monitored at different time intervals at various tool locations during the production process. In 2012, Kuo et al. [2] studied two qualities (power efficiency ratio and slot filling rate error percentage) to analyse and evaluate important parameters in the manufacturing of Fresnel lens solar collectors. In 2014, Sortino et al. [3] compared conventional injection moulding with ICM and VIM to replicate the typical optical thermoplastic microscale prism patterns of Fresnel lenses. The experimental design included a variety of microstructured prism geometries, injection moulding techniques, and specific process parameters. In 2014, Zhuang et al. [4] proposed a conventional Fresnel lens inside and a double total internal reflection (DTIR) lens outside. In order to obtain better irradiation uniformity, a monomorphic algorithm was used to collimate the incident sunlight and optimise the prism displacement based on the principle of different prism focus positions. In addition, the optical performance of the hybrid Fresnel concentrator was verified using a Monte Carlo ray-tracing simulation method. In 2016, Ma et al. [5] used Fresnel’s formula to derive the Stokes reversibility relation, which states that the reflectance on both sides of the interface is equal when light travels along a certain path through two media, regardless of the direction along which it propagates. According to the optimal transmittance condition of prism (OTCP), the transmittance of a curved Fresnel lens can be calculated for a given curvature and focal point, and thus the shape of the lens can be optimised.

As mentioned above, the surface of the Fresnel lens consists of many small V-groove structures that require high precision. By optimising the injection moulding process, the residual stress of the product can be reduced, and the performance can be improved. However, a single genetic algorithm cannot fully consider the constraints of the entire optimisation problem; so, it is necessary to consider the infeasible solutions using thresholding, which increases the workload and solution time and greatly reduces the solution efficiency. At the same time, the particle swarm optimisation algorithm may also lead the results for local optimal solutions. Therefore, we needed methods to improve the solution efficiency and avoid falling into local optimal solutions.

In 2018, Boyaghchi et al. [6] utilized linear Fresnel solar collectors to provide the required energy for a system. A multi-objective optimisation approach based on the fast elite non-dominated ranking genetic algorithm (NSGA-II) was used to find the final optimal thermodynamic, economic, and environmental performances of the system using three decision methods, Shannon entropy, Linear Mapping (LINMAP), and Technique for Order of Preference by Similarity to Ideal Solution (TOPSIS), from the Pareto bound obtained from NSGA-II, combined with the corresponding system design variables. In 2018, Loaldi et al. [7] used a two-dialysis design of experiments (DOE) to investigate the replication ability of IM and ICM on a microstructured Fresnel lens. Laser scanning confocal microscopy was used for the quality control of the optical elements. For this purpose, a detailed uncertainty budget was established for the dimensional measurements of the replicating Fresnel lens, specifically considering the peak and valley step heights and the groove spacing. In 2019, Soni et al. [8] presented a performance analysis of a new optical concentrating solar water heater (OCSWH) using multiple polymethyl methacrylate (PMMA) Fresnel lenses. A series of experiments was conducted by varying key design and operating parameters, such as aperture area, outlet temperature, and mass flow rate, and similar experiments were conducted on a commercially available flat-plate SWH to compare its performance. In 2019, Bensingh et al. [9] reported that injection moulding of double aspheric lenses using polycarbonate materials that minimize the change in volume shrinkage can improve optical quality. Artificial neural networks and particle swarm optimisation techniques were used to optimally predict injection moulding process parameters for double aspheric lenses. The built neural networks were trained and tested using experimental data collected by statistical methods. The well-trained and -tested neural networks were combined with an improved particle swarm algorithm (PSO) to achieve the optimisation of the injection moulding process parameters. In 2019, Roeder et al. [10] investigated a new process chain for the injection moulding of bent microstructured optical elements. A master substrate was prepared using laser direct writing on a curved glass substrate. A subsequent plating process was applied to create a nickel stamped part, and tool inserts for the moulding process were used to show the precise reproduction of the microstructure of the nickel stamped part. Mould inserts were integrated into the injection compression moulding tool to replicate the optical components. In 2020, Ma et al. [11] proposed a new method to simplify the refraction on a prism to obtain the ideal shape of a Fresnel lens. In particular, the ideal curve was closer to its fundamental circle when the total refraction angle was less than 30°, which made it reasonable to use the fundamental circle as the contour line of the Fresnel lens. Then, the change in focal length at the tilted incidence was analysed mainly using simulation and experiments. In 2020, Lightz-Nunez et al. [12] proposed a linear Fresnel reflector optimisation method based on computational fluid dynamics, entropy yield, and evolutionary planning methods. The objective function of the optimisation process considered the maximization of the absorbed radiant solar flux on the receiver tube and the minimization of the total entropy yield.

In summary, non-dominance ranking uses the concept of Pareto optimal solutions to rank individuals in a population so that individuals with a higher non-dominance status are ranked first. This allows the superior individuals to be selected and increases their chances of entering the next generation. Crowding is only compared between individuals in the same dominance hierarchy, and their superiority is determined by calculating the degree of crowding of each individual for each objective function. The elite strategy is to merge the current population with subpopulations generated by selection, crossover, and mutation to compete them to form the next-generation population. This ensures that individuals with better traits remain in the population, increasing the diversity and computational efficiency of the population. The elite strategy also takes care to preserve boundary solutions and thus has an advantage in solving some extremely convex or concave problems by preserving more boundary solutions.

In 2021, Ma et al. [13] proposed an optimal design method for linear Fresnel collector (LFC). First, a mathematical model of the compound parabolic concentrator (CPC) of the LFC was established, and the effects of the choice of the half-receiving angle and the cut-off ratio on the geometric optical efficiency of the CPC were investigated using a ray-tracing method according to the characteristics of the LFC. Second, an unshaded reflector field design method that used the half-reception angle of CPC to limit its aspect ratio was proposed. In 2021, Tan et al. [14] summarized the available fabrication techniques to produce lens structures but did not include moulding methods. In addition, a discussion and analysis of the advantages and future research areas of interest of the reviewed techniques were provided. Advances and further developments in the processing of ultra-precision Fresnel lens structures for advanced optical applications were summarized. In 2021, Men et al. [15] used a combination of a multi-objective genetic algorithm (MOGA) and a Monte Carlo ray tracing (MCRT) method to verify that an efficient objective function can effectively characterize the annual solar flux inhomogeneity of LFRCs. Then, the geometric parameters were combined from each Pareto front using a sequential preference technique known as the method of similarity to an ideal solution (TOPSIS). In 2022, Chang et al. [16] used a Pareto optimisation framework and injection moulding process parameters to optimise the quality of UAV shell parts using a multi-objective optimisation approach. Process parameters, such as melt temperature, filling time, pressure, and pressure time, were used as model variables for the study. The quality of plastic parts was determined by two defect parameters, warpage value and mould index, which were required to be minimal. In 2022, Ahmadpour et al. [17] proposed a new secondary reflecting surface structure consisting of two simple reflecting surfaces. A Monte Carlo ray-tracing (MCRT)-based robustness approach was used to simulate the performance of LFRSC. Then, an improved bait–predator optimisation algorithm (IPPOA) was presented and used for problem optimisation. In 2022, Abbasi et al. [18] proposed a millimetre-wave direction-of-arrival estimation (DoA) technique based on dynamic aperture optimisation. Experiments were shown to limit the radiation in the field of view and to improve the gain of each radiation mode. A method was also proposed to optimise the lens loading dynamic aperture using a mode mixing mechanism controlled by a machine learning-assisted evolutionary algorithm. In 2022, Chang et al. [19] used a multi-objective optimisation approach to optimise the injection moulding defects of automotive pedals. The multi-objective optimal design was mainly used to describe the relationship between cycle time and warpage, and the Pareto boundary was used to identify the cycle time and warpage and determine the deviation function and radial basis-function network. In 2022, Wu et al. [20] proposed a novel biconvex lens compression mould. The parameters were optimised for powder and granular PMMA using the Taguchi method. The surface profile of the finished product was measured, and a factorial response analysis was used to determine the effect of various parameters on the finished product. Compression force was the most important factor affecting powder forming. In 2022, Lin et al. [21] determined the processing conditions for optimising the optical properties and geometric deformation of a plastic Fresnel lens fabricated by injection compression moulding (ICM) technology through mould flow simulations. The analysis process was performed to optimise the optical range difference and optical axial displacement separately using Taguchi’s method. Based on these data, a set of processing parameters that optimised both objectives simultaneously was obtained using the grey correlation analysis. In 2023, Chang et al. [22] applied the LHS method to combine the response surface model with the constraint-generated inverse design network to achieve the multi-objective optimisation of the injection moulding process, shorten the time required to find the optimal process parameters, and improve the production efficiency of plastic parts.

In summary, IMD technology is a technology in which a decorative sheet is placed in an injection mould and the resin is combined with the sheet by injection moulding to form a single unit. This technology can achieve decorative effects during the injection moulding process, making the product both decorative and functional. This paper investigates the performance of the Fresnel lens by comparing IMD technology with conventional injection moulding and observing the effect on residual stress.

According to previous studies, the conventional interpolation technique is prone to edge jaggedness or blurred details when predicting scaling, and it only uses nearby information for modelling and does not use all the information for simulation. In contrast, the Kriging method has local and global statistical properties to analyse and predict the trend of known information. Therefore, in this paper, an improved response surface method based on Kriging is used to calculate the reliability of optimisation by using the Kriging model as a response surface function, and the efficiency and accuracy of the method are verified by arithmetic examples.

The particle swarm optimisation algorithm, which is more suitable for continuous optimisation problems and is usually used in combination with neural networks to achieve good optimisation results, has been used in previous studies. However, for problems that require simultaneous optimisation of multiple objectives, the particle swarm optimisation algorithm performs poorly, exhibiting poor local search capability, a tendency to fall into local extremes, and low search accuracy. On the other hand, although the method of using a genetic algorithm to optimise the injection moulding process meets the optimisation requirements of the Fresnel lens, the ordinary genetic algorithm has a long computation time and requires several iterations to achieve the desired results.

Based on the previous research results, this paper combines the NSGA-II algorithm and the Kriging prediction model to determine the optimal values of process parameters so as to effectively reduce the residual stress of the product. The NSGA-II algorithm can obtain uniformly distributed Pareto bounds with good convergence and robustness. The fitting results of the Pareto boundaries show that there is no significant trade-off between dent and volume shrinkage or dent and warpage. In this paper, a non-explicit genetic algorithm model is developed for the complex relationship between nonlinear mapping design parameters and residual stresses. By combining the genetic algorithm and the prediction model, the optimal values of the process parameters are found to reduce the mould index and obtain the corresponding residual stress values. The framework is shown in Figure 2.

## 3. Material

The field of optical polymers included a variety of different materials that had glass-like properties. Optical polymers can generally be divided into two different categories. The first category consists of thermosetting or thermoplastic resins, such as polycarbonate (PC), and the second category is polymethylmethacrylate (PMMA), also known as acrylic or plexiglass. As with other polymers, the specific properties of optical polymers can be adapted and tailored to the needs of each application.

However, among the many optical polymers, the Fresnel lens is a special design that combines the advantages of polymers with a specific optical morphology. By carefully selecting and adapting optical polymer materials, Fresnel lenses are able to function in a diverse range of optics, providing a unique option for creating more precise, lightweight, and powerful optical systems.

In addition to the materials mentioned above, there is a new material known as cyclic olefin polymer (COP). This polymeric material is widely used in optics due to its low water absorption, transparency, and optical properties similar to PMMA. Table 1 shows the physical properties of three polymers (PC, PMMA, and COP). Figure 3 shows the pressure–volume–temperature (PVT) comparison of these three materials. In terms of density, COP is lighter than PMMA. Conversely, COP has a higher viscosity at all selected melting temperatures. The slope of the curve indicates that the viscosity decreases more uniformly with an increasing shear rate. In contrast, the viscosity of PMMA remains constant until the shear rate reaches 100 1/s, at which point the viscosity converges to the same value regardless of melt temperature.

Based on the comparison above of the properties of the three materials, COP and PC are more commonly used in other applications, although they can also be used to make Fresnel lenses. COP has a lower softening temperature and a lower coefficient of thermal expansion for specific optical applications. PC has a higher softening temperature and higher compressive modulus for applications that require greater stiffness and high temperature resistance. However, it can be more expensive and less optically transparent than PMMA. In summary, PMMA is a more suitable choice for the manufacture of Fresnel lenses based on optical clarity, processability, impact resistance, and cost effectiveness.

The PVT curve of PMMA is shown in Figure 3, with a glass transition temperature (Tg) of approximately 105–110 °C. The linear coefficient of thermal expansion of PMMA is about 70–90 × 10^−6^/°C, which means that as the temperature changes, the volume of the PMMA material changes accordingly. The compressive modulus of PMMA is usually in the range of 2–3 GPa. The compressive modulus reflects the compressive performance of the material under pressure, and PMMA performs well in this respect.

## 4. Methods

(1)In this study, a suitable injection moulding process was selected for the investigation. Traditional optimisation methods require a large number of numerical simulation calculations, which are less efficient. In this paper, a Kriging model was proposed to map the approximate functional relationship between the process parameters and optimisation objectives. Furthermore, the main factors affecting the Fresnel lens node displacement and residual stress were taken as design variables to determine their value ranges.(2)In order to achieve the accurate optimisation of the injection process parameters, a non-dominated genetic algorithm (NSGA-II) was proposed in this paper to perform multi-objective optimisation based on previous work. The balanced Pareto optimal solution set was obtained by the selection, crossover, and variation operations of the NSGA-II algorithm. After the experimental validation, a suitable combination of parameters in the Pareto optimal solution set was selected for practical injection experiments to verify the validity of the optimisation results.

Specifically, in this paper, the Kriging model was constructed using the relevant parameters that affected the accuracy of the model, and a high-accuracy model was selected. The model was optimised by combining with the NSGA-II algorithm to obtain the Pareto optimal solution set, so as to obtain the best injection moulding parameters, optimise the displacement offset of the Fresnel lens, and reduce the residual stress, thus improving the transmission rate and performance of the lens.

### 4.1. In-Mould Electronic Decoration (IMD) Technology

This paper studied the injection moulding of Fresnel lenses, mainly by comparing the IMD, injection compression moulding, and two-stage injection moulding technologies; Table 2 mainly describes the advantages and disadvantages of the three methods and their application areas.

IMD technology, or in-mould decoration, is a novel in-mould decoration technique that combines injection moulding and decoration. It integrates functionality and aesthetics, making it widely used in the injection moulding process. IMD is a globally popular surface decoration technique that involves applying a cured transparent film to the surface, followed by a printed pattern layer and an injection layer, with a colour layer in between. This design prevents surface scratching, improves rub resistance, and maintains vibrant colours over time. The coating on the surface of optical lenses also protects their microstructure.

Injection compression moulding (ICM) combines the techniques of injection and compression moulding. In this process, the mould is not completely closed. As the molten material is injected, the locking mechanism begins to operate, gradually closing the mould. At the end of the process, the mould is fully closed and the product shape is formed. Typically, position control and pressure control modes are used to control mould position. The main advantage of injection compression moulding is improved dimensional stability at lower injection pressures during filling. Uniform pressure distribution within the mould cavity results in a better product yield and improved residual stresses, warpage, volume variation, and birefringence. Injection compression moulding is commonly used for optical components and thin-profile products.

In the two-stage injection moulding process, the lens is axially symmetrical, and thus there is no need for angular alignment. In two-stage injection moulding, the first shot is injected at a low speed, with the nozzle located away from the cold sprue, and then the speed is increased for the second shot to fill the cavity. The aim is to reduce the flow time of the plastic from the gate to the end point and to maintain the viscosity of the material in the filling stage at a minimum solidification state. However, it is difficult to control the correct switchover point for the holding pressure at high injection speeds. Therefore, a multi-stage decrement approach must be used to effectively control the switchover point. Two-stage injection moulding can significantly reduce cooling time and save injection time, and thus can improve the efficiency of the large-scale production of plastic parts.

In this study, we compared three moulding methods, namely, IMD moulding, two-stage injection moulding, and injection compression moulding, to determine which method minimises the displacement of V-groove nodes on the surface of the Fresnel lens. We performed simulation analyses using these three methods and selected 10 nodes to compare the displacement produced by each injection moulding method, as shown in Table 3.

Based on the results in Table 3, it can be observed that the two-stage injection moulding process results in the largest node displacement for the Fresnel lens, indicating its poor performance. The average displacement of the surface nodes was 0.861 mm. On the other hand, the injection compression technique produced the smallest average node displacement of 0.251 mm. The IMD technique produced an average node displacement of 0.659 mm. It is worth noting that the injection compression method produces favourable results and is suitable for regular flat optical lenses. When the lens surface is smooth, the use of injection compression can produce higher-quality products. However, in the case of the Fresnel lens studied in this paper, which has a special serrated structure, the injection compression process may cause damage to the surface microstructure. In addition, Fresnel lenses are commonly used to capture infrared light, and the addition of an IMD film can improve their performance. Therefore, in this study, the IMD technique was primarily used to manufacture the product, with the injection moulding parameters optimised to achieve better nodal displacement results.

### 4.2. Kriging Surrogate Model

The Kriging proxy model is a generalised regression model based on a stochastic process. It can predict the unknown sample points by using the known sample points, while using the change in variance to represent the change in space. It can guarantee the minimum estimation error obtained by the spatial distribution and then construct a response value, which cannot be obtained by a simulation analysis. It has the characteristics of better local estimation and higher accuracy simulation for non-linear problems and is mostly used in engineering application analysis.

The input variables and response values of m observation points are known, denoted as $S_{m}=\left[x_{1},x_{2},\ldots ,x_{m}\right]$ and $Y={\left[y_{1},y_{2},\ldots ,y_{m}\right]}^{T}$, where $x_{i}\in R^{n}$,$\;y_{i}\in R$, i=1,2,…m. It is necessary to interpolate and analyse the response value *y* of unknown x. The relation expression between input variables and response values in the Kriging model is written as Equation (1):(1)$$y\left(x\right)=F\left(\beta ,x\right)+z\left(x\right)=f^{T}\left(x\right)\beta +z\left(x\right)$$

M and n are the dimensions of design variable x. F(β,x) in Equation (1) is a linear regression model of data, and the number of regression vectors reflects the change in the mean value in the fitting process of the model. The linear regression model p=1, $f_{1}\left(x\right)=1$ is used in practical engineering applications. z(x) is a stochastic process model, established by data observation and the quantification of data correlation, whose mean value is θ and variance is $\sigma^{2}$. The covariance between sampling points is shown in Equation (2):(2)$$\{\begin{array}{c} E\left\{z\left(x\right)\right\}=0\\ Var\left[z\left(x\right)\right]=\sigma^{2}\\ Cov\left[z\left(x_{i}\right),z\left(x_{j}\right)\right]=\sigma^{2}\left[R\left(\theta ,x_{i},x_{j}\right)\right] \end{array}$$

The mathematical expression of R(θ,w,x) is shown in Equation (3), where θ is a key parameter of the Gaussian correlation function. The correlation between sampling points can be adjusted adaptively by optimising θ.
(3)$$R\left(\theta ,w,x\right)=\overset{n}{\underset{i=1}{\prod }}R_{i}\left(\theta_{i},w_{i}-x_{i}\right)$$

The spatial correlation function controls the smoothness of the Kriging model, the interrelationship of nearby points, and the correlation between quantified observations. Therefore, the Kriging model can treat any response value as a random variable subject to normal distribution, so that the model is not limited to a specific form and has strong flexibility.

The standard Kriging model has seven commonly used spatial correlation functions: exponential, exponential Gaussian, Gaussian, linear, spherical, cubic, and spline function models. Among them, the Gaussian model is shown in Equation (4). It provides a relatively smooth and infinitely differentiable surface and is used in a large number of practical analytical applications.
(4)$$R_{i}\left(\theta_{i},d_{i}\right)=\exp\left(-\theta_{i}{\left|d_{i}\right|}^{2}\right)$$

Using the function related to type R to determine the distance between two points, with $\left|w_{i}-x_{i}\right|,$ the maximum likelihood estimation theory and unconstrained optimisation methods can be used to maximize the type (see Equation 5). The optimal value of θ can be calculated as follows:(5)$$-\left(mln\sigma^{2}+ln\left|R\right|\right)/2$$

In practical situations, although the optimal value θ does not necessarily lead to an optimal approximation curve, the quality of the approximation curve improves when the approximation result is close to the optimal solution. In this thesis, a Kriging agent model was constructed and optimised using MATLAB software, which provided functions for the experimental design, model building, and computation to facilitate the construction of the Kriging agent model and the optimisation computation.

### 4.3. Multi-Objective Genetic Algorithm for Non-Dominated Sorting (NSGA-II)

The non-dominated sorting genetic algorithm (NSGA-II) is a multi-objective optimisation algorithm that extends the traditional genetic algorithm. In addition to the conventional genetic operators of selection, crossover, and mutation, this algorithm introduces a new selection mechanism. The rapid sorting of non-dominated solutions, crowding distance, and elitism are integral parts of NSGA-II. The main steps of NSGA-II are outlined below and shown in Figure 4.

Unlike traditional genetic algorithms, which consider the principle of elitism in selection, NSGA-II incorporates the concept of crowding distance to ensure diversity among solutions, with a focus on non-dominated sorting methods. The key advantage of the NSGA-II optimisation process is that individuals in the population are ranked according to the degree to which they are dominated by others. This allows for the rapid identification of non-dominated individuals in the population, effectively preserving the superior individuals. By using a crowding distance operator, NSGA-II preserves the diversity of the population, resulting in a more evenly distributed set of individuals and improving the global search capability of the algorithm.

Step 1: Initial population: Randomly generate a specified number of individuals to form the initial population.Step 2: Fitness evaluation: Calculate the fitness score and non-dominated ranking for each individual.Step 3: Crowding distance calculation: Calculate the crowding distance for each individual to maintain diversity in the population, representing the density around each individual.Step 4: Non-dominated sorting: Divide the individuals in the population into several levels. The first level consists of non-dominated solutions, i.e., solutions that are not inferior to any other individual with respect to all objective functions.Step 5: Selection of individuals: Select individuals based on their non-dominated ranking and crowding distance. Give priority to individuals with higher ranking and larger crowding distance.Step 6: Crossover and mutation: Perform crossover and mutation operations on the selected individuals to generate new offspring.Step 7: Update the population: Add the newly generated offspring to the population, replacing the weaker individuals to form the next generation population.Step 8: Check termination condition: If the termination condition is met, stop the algorithm and output the non-dominated solution set as the final solution set. Otherwise, return to step 2.

In the optimisation process of NSGA-II, the fitness value and the non-dominated ranking of individuals are crucial. By calculating and updating these values, the algorithm progressively discovers a set of non-dominated solutions that assist the decision maker in making multi-objective optimisation decisions. In addition, NSGA-II takes into account the diversity of individuals during selection by using the crowding distance to maintain the diversity of the population. This allows the algorithm to explore the solution space more effectively and avoid becoming trapped in local optima.

## 5. Case Study

In this study, we used IMD technology in combination with PMMA material to produce Fresnel lenses and investigated the changes in nodal displacement and residual stress under different combinations of injection moulding parameters. IMD technology is an advanced technique that allows the direct embedding of decorative patterns into injection-moulded products, achieving seamless integration of the pattern with the lens surface during the manufacturing process. We chose PMMA as the base material because of its excellent optical properties, high transparency, and good processability, making it highly suitable for the manufacture of Fresnel lenses. By using IMD technology, we were able to embed intricate patterns directly onto the lens surface, achieving both decorative and functional integration. By combining the PMMA material with the IMD technology, our aim was to investigate the optimum parameter combination for the lenses and the average transmittance that the lenses could achieve.

During the production process of Fresnel lenses, problems such as internal residual stress, birefringence, and surface accuracy are often encountered. If the mould designers and process engineers can determine the shear rate, temperature, pressure, and time of the molten material flowing through the gating system and mould cavity in advance, it is possible to reasonably determine the mould structure and injection process, thereby improving the first-pass yield of the products. Therefore, in this study, we used mould flow software for the pre-simulation to observe the target values of node displacement and residual stress obtained. In this section, the Kriging prediction model was combined with the NSGA-II algorithm to determine the optimal combination of injection parameters to improve the quality and optical performance of the lenses.

First, the initial combination of injection parameters was determined based on the characteristics of the PMMA material. Then, simulation experiments were performed on the injection process of Fresnel lenses to observe the influence of different injection parameters on the quality of the moulded parts (mainly the magnitude of residual stress and displacement). Response surface plots of the three moulding parameters, namely, holding pressure, melt temperature, and holding time, were obtained, as shown in Figure 5.

According to the results in Figure 5, the effects of holding pressure and melt temperature on the comprehensive index are very significant. Therefore, in order to reduce the number of experiments in the actual experimental process, we focused on the influence law of these two factors on the moulding quality of Fresnel lenses and considered the factors of holding pressure time, injection rate, and cooling time as insignificant influencing factors. After the experimental verification and analysis, we found that the parameter settings of holding pressure and melting temperature both affected the nodal displacement and residual stress of the plastic part, thus ensuring the geometric accuracy of the Fresnel lens and avoiding the occurrence of low transmittance and birefringence. Therefore, in this paper, we chose to optimise the two injection parameters of holding pressure and melting temperature.

When optimising the holding pressure, a pressure too high can lead to warpage and deformation, flying edges, and even mould rise. This is because too much melt is compressed into the mould cavity, and the volume reduction caused by cooling shrinkage cannot be fully compensated for. At the end of cooling, the pressure in the mould cavity does not return to zero, resulting in residual stresses in the moulded product. In the Fresnel lens manufacturing process, the residual stress affects the performance of the lens, including transmittance and geometric accuracy; so, it is crucial to optimise the holding pressure.

With regard to the optimisation of the melt temperature, in general, the viscosity of the polymer is inversely proportional to the temperature when the temperature is higher than the viscous flow temperature. As the temperature increases, the free volume of the melt increases, the intermolecular interaction force weakens, the fluidity of the polymer increases, and the melt viscosity decreases exponentially. In polymer injection moulding, temperature control is the main means of adjusting viscosity to improve the melt filling capacity. Therefore, as the material temperature increases, the melt viscosity decreases. If the injection and holding pressures are kept constant at this time, the gate freezing rate is slower, increasing the holding time and the effect of compensatory shrinkage, which in turn increases the density and reduces shrinkage. This also has a greater effect on the geometric accuracy of the Fresnel lens.

### 5.1. The Establishment of the Kriging Model

During the experiment, we set up two modal data by designing 20 sets of sampling points with holding pressure and melting temperature. We simulated the experimental data using modal flow software and present them in Table 4, which includes the output responses of nodal displacements and residual stresses on the lens surface.

To predict the unknown sample points, we used the Kriging proxy model, which is a generalised regression model based on a stochastic process. The Kriging model can effectively predict the unknown sample points using known sample points and represents the spatial variation through variance changes. First, we calculated the weight coefficients of the sample points; then, we performed a linear fit to the sample points and the weight coefficients and used interpolation calculations to predict the values of the desired points.

We imported the data in Table 4 into the MATLAB software and selected the Kriging model in the Toolbox module. We then imported the parameters and response values and fit the model using a Gaussian spatial correlation function. We then ran the simulation of the Kriging model and checked the accuracy. Figure 6 shows the Kriging prediction model with regard to nodal displacements, while Figure 7 shows the Kriging prediction model with regard to residual stresses. See Figure 6 and Figure 7.

Based on the results in Figure 6 and Figure 7, we found four minimum peaks in the Kriging prediction model using the influenced nodal displacements. By comparison, we found that the minimum value of the nodal displacement was 0.5381 mm. In this case, the set value of the holding pressure was 304.4 MPa and the set value of the melting temperature was 252.3 °C.

For the Kriging prediction model of residual stresses, we found three minimum peaks. At the minimum peak, the minimum value of residual stress was 0.1812 MPa. At this point, the set value of the holding pressure was 302.4 MPa, and the set value of the melt temperature was 255.7 °C.

Then, we used the NSGA-II algorithm to find the optimal solution and compared and contrasted the results obtained with the Kriging prediction model.

### 5.2. NSGA-II Multi-Objective Optimisation

After verifying the effectiveness of the response surface model, the selected NSGA-II genetic algorithm was used to find the optimal Pareto solutions for the two mass objectives of node displacement, $R_{1},$ and residual stress force, $R_{2}$. For the optimisation of these two quality objectives, the following objective functions and constraints were established:(6)$$F=Min\left\{M_{1},M_{2}\right\}$$

Using the Genetic Algorithm Toolbox provided with the MATLAB software, we selected the NSGA-II algorithm and converted the established second-order response surface model into an objective function file. We used the following data settings: the number of variables was 2, the holding pressure range was 180 to 330 MPa, the melt temperature range was 220 to 260 °C, the population size was set to 50, the crossover rate was 80%, the variation rate was 9%, and the number of terminating generations was 200. The deviation of the fitness function value was set to 1 × 10^−100^.

By performing the above settings and running the genetic algorithm, we obtained the Pareto solution sets. These solution sets are shown in Figure 8.

Based on the results in Figure 8, we selected the optimal solution with an average node displacement of 1.04 mm and a residual stress of 0.04328 Mpa as the objective of the multi-objective optimisation. In this solution, the holding pressure was set to 320.35 Mpa and the melting temperature was set to 251.40 °C.

To verify the reasonableness of the results, we ran simulations using this set of parameters and obtained the results shown in Figure 9.

Based on the analysis of the data in Figure 9, we can conclude that the average value of the nodal displacement is about 0.675 mm and the average value of the residual stress is about 0.075 Mpa when the simulation is performed using parameters set to a holding pressure of 320.35 Mpa and a melting temperature of 251.40 °C. This is similar to the results obtained by iteration of the NSGA-II algorithm and is also consistent with the results of the Kriging prediction model.

Therefore, based on the simulation results, it can be concluded that it is reasonable to select the holding pressure of 320.35 Mpa and the melting temperature of 251.40°C as the parameter settings that can achieve the optimisation of nodal displacement and residual stresses.

## 6. Results and Discussion

In this study, a Fresnel optical lens made of PMMA polymer was manufactured by compression and injection moulding. However, the manufacturing of precision Fresnel optical lenses faces challenges in reducing residual stresses and improving the quality of surface microstructures to achieve precise replication. In this research, the NSGA-II algorithm was used to optimise process parameters based on multiple quality objectives. The optimal process parameters determined were a holding pressure of 320.35 Mpa and a melting temperature of 251.4 °C.

In this section, the distribution of nodal displacement and residual stress were obtained through simulations using the optimal parameter combination. The final product was produced using an all-electric injection moulding machine, and the transmittance was tested using a Lambda 950 UV/Vis spectrophotometer.

By combining the Kriging prediction models and the NSGA-II multi-objective optimisation algorithm, the best combination of parameters was obtained: a holding pressure of 320.35 Mpa and a melting temperature of 251.40 °C. We performed simulations and analyses using this parameter set, and the corresponding node displacement is shown in Figure 10, while the residual stress is shown in Figure 11. The node displacement obtained from the original solution was compared with the optimised node displacement, as shown in Table 5.

In summary, we can conclude that increasing the holding pressure and melting temperature can achieve a better optimisation effect, and the optimisation efficiency can reach 59.64%. By increasing the holding pressure and melting temperature, we can avoid the displacement changes of the surface nodes of the plastic parts and reduce the moulding shrinkage of the plastic parts, thus preventing the deformation of the film surface caused by the shrinkage difference.

Transmittance is one of the most important indicators of the quality of the finished Fresnel lens. Transmittance indicates the degree of light loss through the object within the object, i.e., the ratio of transmitted light to the original incident light. In the application of Fresnel lenses, high transmittance means that less light is lost through the lens, allowing a more efficient transmission and focusing of light energy.

In this paper, the optimum injection parameters were obtained by optimising the design and controlling the injection process parameters. The transmittance of the Fresnel lens was measured using a Lambda 950 UV/Vis spectrometer, as shown in Figure 12, where the horizontal coordinate was the infrared wavelength and the vertical coordinate was the transmittance of the Fresnel lens at that wavelength. By compiling the data, we found that the transmittance of the Fresnel lens could reach an average of 95.53% in a given wavelength range.

High transmittance means that Fresnel lenses can effectively transmit infrared light and perform well in applications such as infrared temperature measurements, infrared thermal imaging, infrared communications, and infrared sensors. The increased transmittance indicated that the manufacturing quality of Fresnel lenses was effectively controlled and optimised to meet the requirements of specific applications and improve the efficiency and performance of the system. Therefore, a high transmittance value can be one of the most important indicators of the excellent quality of the finished Fresnel lens, reflecting the superior performance of the lens in terms of optical transmission and an optimised manufacturing process.

As can be seen from Figure 12, the Fresnel lens is very good at transmitting near-infrared light. Therefore, Fresnel lenses can be used in many applications. Some examples of possible applications are as follows:Infrared thermometry instruments: Fresnel lenses can be used to collect infrared radiation energy and transmit it to a thermometer for temperature measurement.Infrared thermal imagers: Fresnel lenses can be used to focus infrared radiation to produce images of the temperature distribution of a sample.Infrared communication system: Fresnel lenses can focus the emitted infrared signal on the receiver to improve the communication distance and stability.Infrared sensors: Fresnel lenses can be used to collect infrared radiation energy and transmit it to the sensor to detect infrared radiation signals from the target.

In summary, Fresnel lenses have a wide range of infrared applications, including infrared temperature measurement, thermal imaging, communications, and sensing. These applications demonstrate the importance and versatility of Fresnel lenses in optical devices.

## 7. Conclusions

There are many critical factors to consider when controlling the quality of injection-moulded products; however, they are not completely uncontrollable. Factors, such as plastic flow, heat transfer, and mould temperature during the injection moulding process, can affect the quality of injection-moulded products. In order to improve the quality of injection-moulded products, these factors must be designed and controlled in a comprehensive and optimal way. In this study, the two parameters of holding pressure and melting temperature were optimised, and their effects on the nodal displacement and residual stress of Fresnel lenses were studied to provide a reference for the quality control of Fresnel lenses.

Through the study of different plastic melt properties (holding pressure and melting temperature), we explored the relationship between moulding parameters and nodal displacement as well as surface residual stress, thus providing a method for optimising injection moulding parameters and improving product quality. Through 20 sets of test specimens with different injection parameters, we obtained the response values of node displacement and residual stress. Then, the Kriging model was used to predict these 20 sets of sample points; the prediction model was obtained and the values of injection moulding parameters that minimised the response values were determined.

We then optimised the injection parameters using the non-dominated genetic algorithm (NSGA-II) to obtain the optimal injection parameters for the Pareto boundary, i.e., the Fresnel lens. Experimental validation was performed to verify that the effect of these optimal parameters was as expected. Finally, we tested the transmittance using a Lambda 950 UV/Vis spectrometer. The main conclusions drawn from the results of the study are as follows:The optimised target results were obtained for two parameters, hold time and melt temperature, through 20 sets of test samples with different injection parameter settings. The optimum process parameters were 320.35 MPa holding pressure and 251.40 °C melt temperature. Under these conditions, the transmittance can reach 95.43% and the optimisation rate of node displacement can reach 59.64%.The experimental results show that there is a relationship between the effects of holding pressure and melt temperature on the nodal displacements and the average residual stress. The holding pressure is the main influencing factor, followed by the melt temperature. Therefore, the ranking of the influencing factors is holding pressure > melt temperature.By comparing three injection moulding methods, including IMD (injection moulding combined with electronic decoration technology), injection compression, and two-stage injection moulding Fresnel lens processing, the experimental results show that the average level of node displacement change is the level when using IMD technology, which is 0.628 mm, while the average level of node displacement change is the greatest when using two-stage injection moulding, which is 0.816 mm.The optimisation of injection process parameters is equivalent to a black-box optimisation problem, and the NSGA-II algorithm is a good choice for solving such problems. By using the elite strategy, the convergence can be accelerated, the inferior solutions can be rapidly eliminated, the number of iterations can be reduced, and the optimisation can be accelerated. The algorithm is able to find better solutions and converge better near the Pareto optimal frontier.

In summary, this study investigated the effects of holding pressure and melting temperature parameters on the nodal displacement and residual stress of Fresnel lenses by optimising them. The optimal injection parameters were obtained using a multi-objective optimisation algorithm and experimental validation, and the transmittance was tested by using a Lambda 950 UV/Vis spectrometer. The results show that the performance of the Fresnel lens is significantly improved with the optimal injection parameters. This has important implications for the manufacturing and quality control of Fresnel lenses.

## Figures and Tables

**Figure 1 polymers-15-03403-f001:**
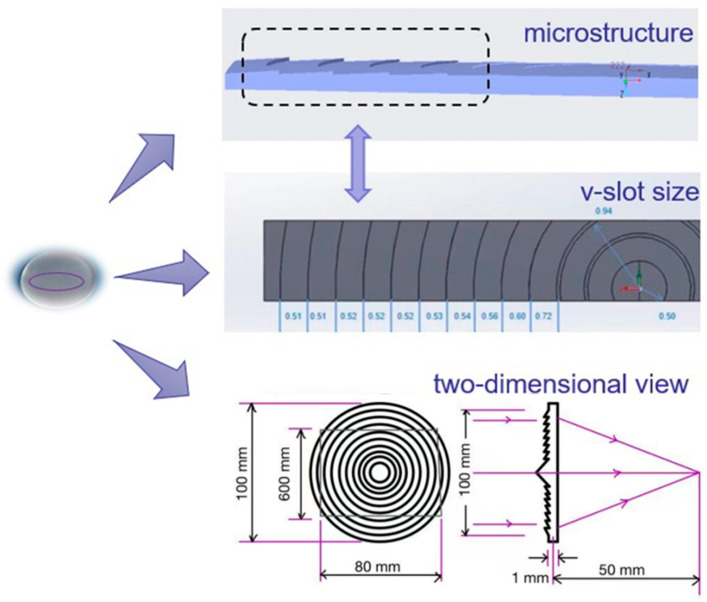
Fresnel lens profile.

**Figure 2 polymers-15-03403-f002:**
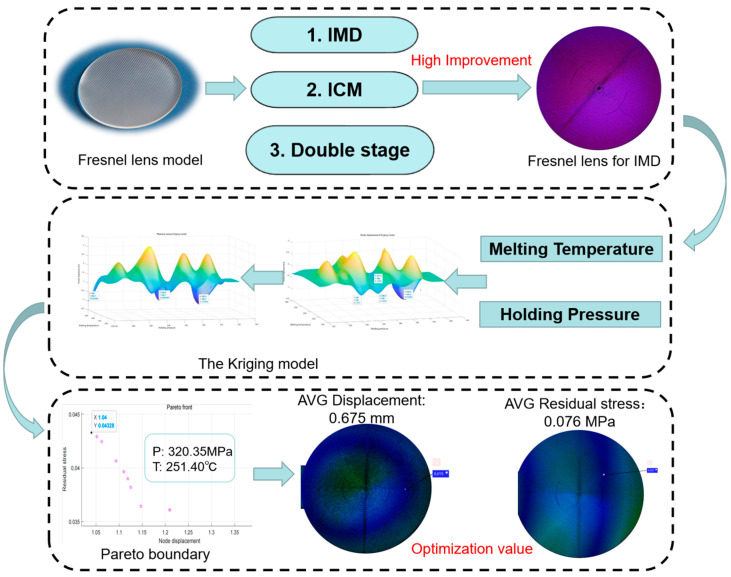
NSGA-II optimisation concept for Fresnel lens experiments.

**Figure 3 polymers-15-03403-f003:**
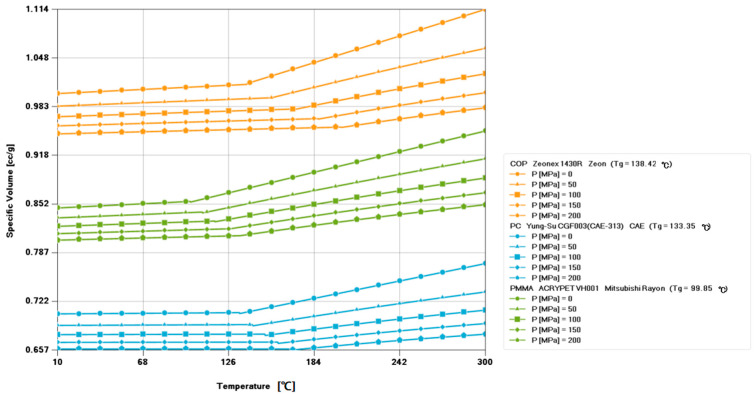
PVT comparison of PC, PMMA, and COP materials.

**Figure 4 polymers-15-03403-f004:**
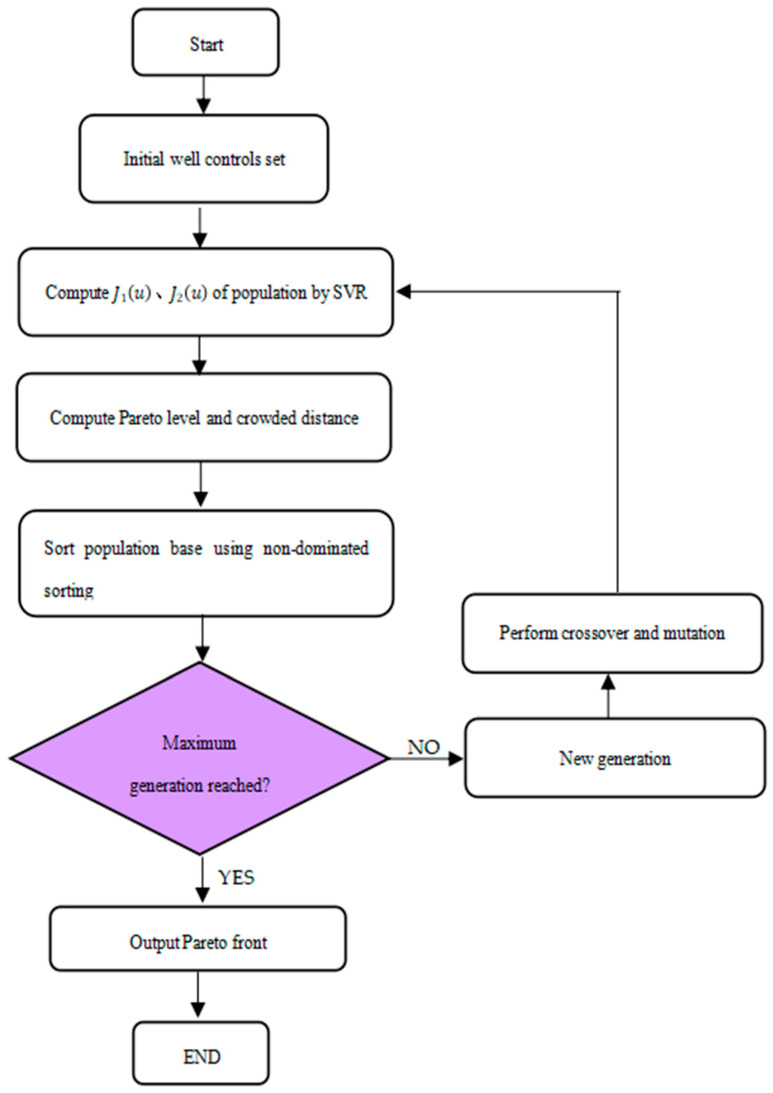
Flowchart of NSGA-II calculation.

**Figure 5 polymers-15-03403-f005:**
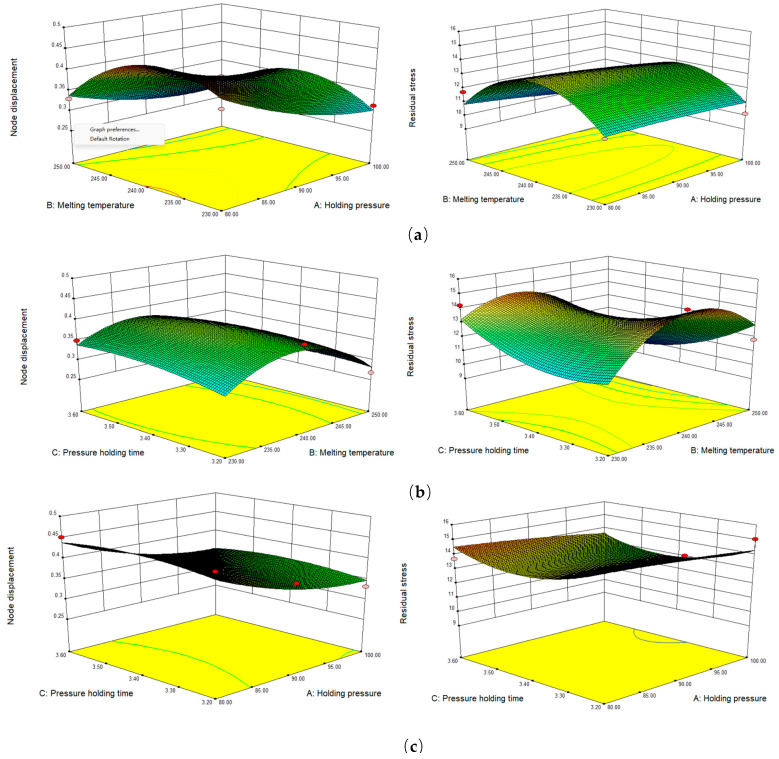
Response surface diagram of three factors. (**a**) shows the influence of holding pressure and melting temperature on the displacement and residual stress of the joint, respectively. (**b**) shows the influence of pressure holding time and melting temperature on node displacement and residual stress, respectively. (**c**) shows the influence of pressure holding time and holding pressure on the displacement and residual stress of the node, respectively.

**Figure 6 polymers-15-03403-f006:**
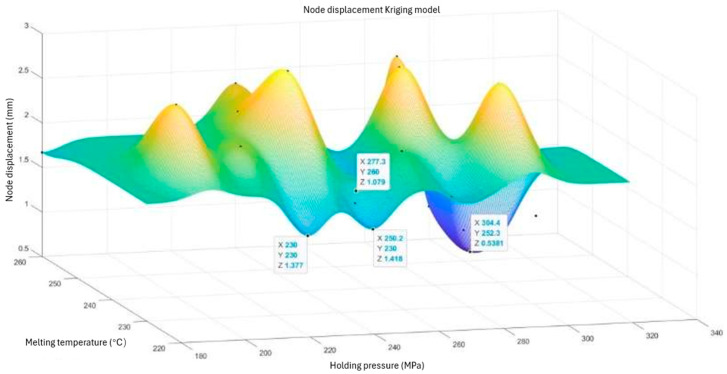
Node-displacement Kriging model.

**Figure 7 polymers-15-03403-f007:**
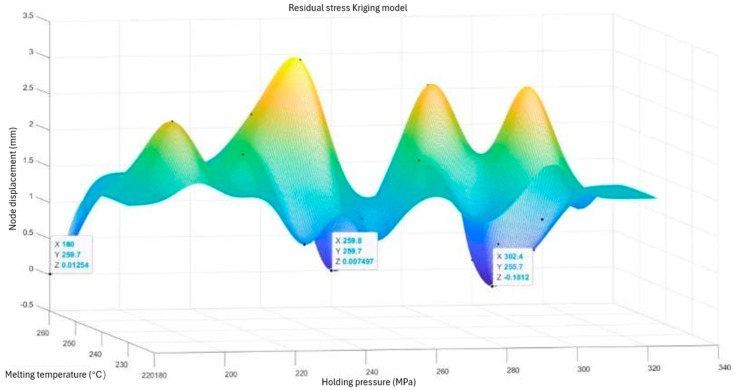
Residual stress Kriging model.

**Figure 8 polymers-15-03403-f008:**
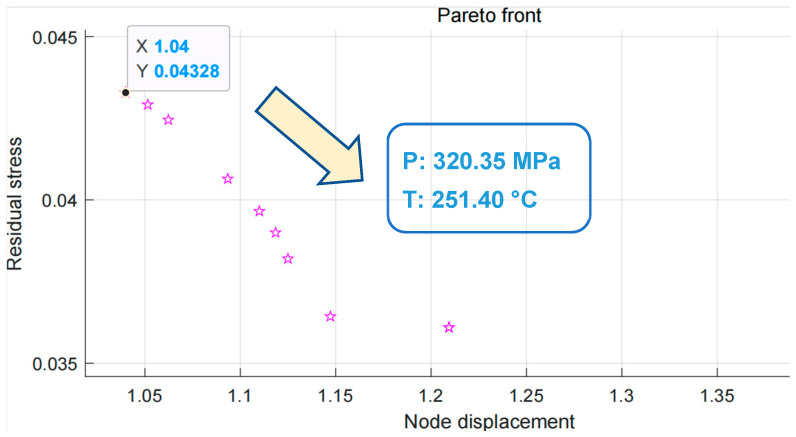
Pareto boundary optimal solution set.

**Figure 9 polymers-15-03403-f009:**
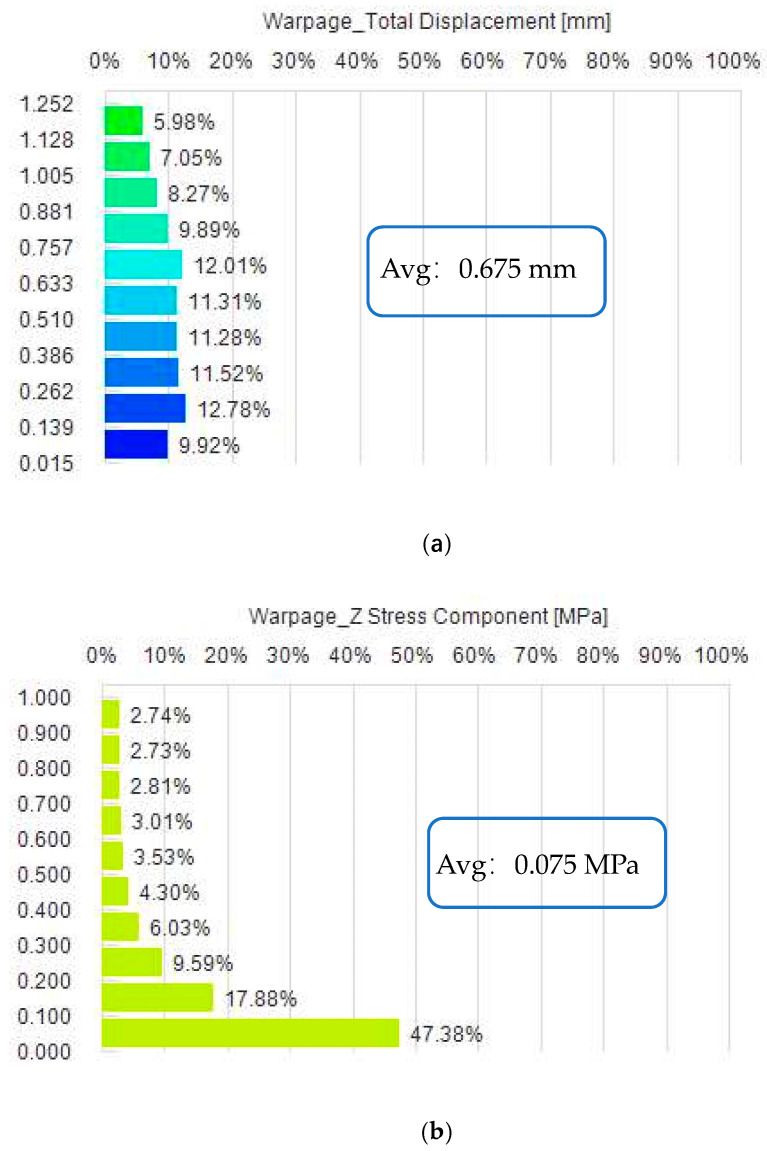
Verification of Pareto results. (**a**) shows the distribution ratio of Fresnel lens displacement. (**b**) shows the proportion of residual stress distribution in the Fresnel lens.

**Figure 10 polymers-15-03403-f010:**
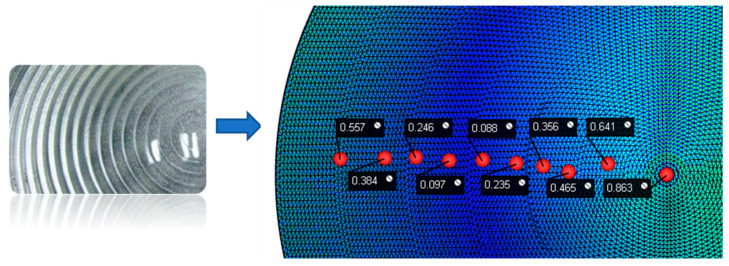
Optimal injection parameter node displacement.

**Figure 11 polymers-15-03403-f011:**
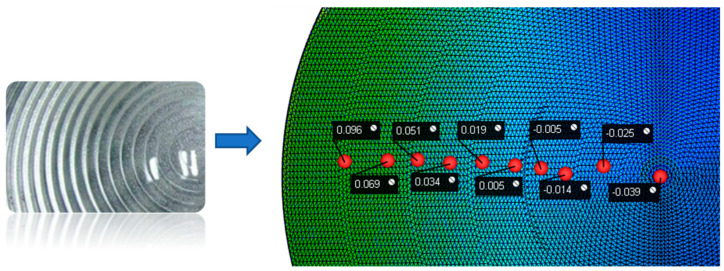
Optimal injection residual stress.

**Figure 12 polymers-15-03403-f012:**
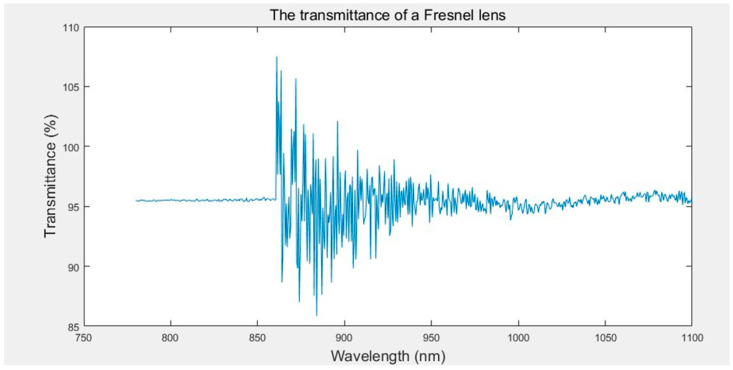
The transmittance of a Fresnel lens in near-infrared light.

**Table 1 polymers-15-03403-t001:** Comparison of properties of three materials.

Material	Main Application	Merits	Shortcomings
Polycarbonate (PC)	Glass assemblies, automotive and electronics, appliances	Good toughness and impact strength	Low thermal deformation temperature, poor weather resistance
Polymethyl Methacrylate (PMMA)	Optical glass, optical fibre, all types of instrument dials	Light transmittance and weather resistance	Brittle material, low hardness, easily faded
Cyclo Olefin Polymer (COP)	Lenses and screens of medical optical components and electronic products	Low birefringence, low water absorption, high rigidity, high heat resistance	High unit price, recycled materials cannot be used

**Table 2 polymers-15-03403-t002:** Comparison of three injection moulding methods.

Method	Advantages	Disadvantages	Applications
IMD	Scratch resistance, solvent resistance, long service life	Injection moulding, temperature, pressure, speed is difficult to control	Automobile instruments, IC decorative cases
ICM	Reduces injection pressure and residual stress	Long cycle, low production efficiency	Thin-walled, miniaturized parts, optical lenses
Two-stage	Different plastics integrated into one, higher quality, better strength	Compatibility of the two materials needs to be considered	Handles, headlights, phone buttons

**Table 3 polymers-15-03403-t003:** Node displacement under three injection moulding methods.

Number of Nodes	IMD (mm)	ICM(mm)	Two-Stage Injection Moulding (mm)
1	0.859	0.416	1.580
2	0.551	0.306	1.132
3	0.301	0.188	0.511
4	0.269	0.096	0.120
5	0.272	0.073	0.325
6	0.609	0.151	0.595
7	0.654	0.229	0.895
8	0.842	0.294	1.096
9	0.976	0.353	1.190
10	1.255	0.400	1.166
Average	0.659	0.251	0.861

**Table 4 polymers-15-03403-t004:** Sample data set and corresponding response results.

Node	Holding Pressure (MPa)	Melting Temperature (°C)	Displacement Offset (mm)	Residual Stress (MPa)
1	300	260	0.870	0.122
2	280	250	2.700	2.695
3	290	260	2.563	2.433
4	290	220	1.745	1.276
5	240	260	2.343	2.077
6	280	260	1.163	1.054
7	180	260	1.669	0.011
8	260	260	1.537	0.010
9	330	260	1.231	0.818
10	310	250	0.706	0.399
11	300	300	0.847	0.492
12	260	260	1.331	0.783
13	230	250	2.287	2.325
14	270	240	2.015	1.807
15	250	235	1.586	1.089
16	280	235	1.608	1.117
17	250	230	1.418	0.860
18	230	230	1.377	0.820
19	220	240	2.150	1.821
20	220	240	2.625	2.394

**Table 5 polymers-15-03403-t005:** Comparison of displacement before and after optimisation.

Node	Displacement (mm)	Optimised Displacement (mm)	Optimised Efficiency
1	0.859	0.557	64.84%
2	0.551	0.384	69.69%
3	0.301	0.246	81.73%
4	0.269	0.097	36.06%
5	0.272	0.088	32.35%
6	0.609	0.235	38.59%
7	0.654	0.356	54.43%
8	0.842	0.465	55.23%
9	0.976	0.641	65.68%
10	1.255	0.863	68.76%
Average	0.659	0.393	59.64%

## Data Availability

Not applicable.

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
