# Peer review of "Application of the NSGA-II Algorithm and Kriging Model to Optimise the Process Parameters for the Improvement of the Quality of Fresnel Lenses"

_polymers, 2023, doi:10.3390/polym15163403_

Round 1

Reviewer 1 Report (Previous Reviewer 1)

accept

sufficient quality of English language

Author Response

Please see enclosed, thank you.

Reviewer 2 Report (Previous Reviewer 2)

Authors implemented the suggested comments and advices.

No important issues detected

Author Response

Please see enclosed, thank you.

Reviewer 3 Report (Previous Reviewer 3)

English was improved. Minor revision is required. See my comments below.

1. Title should be revisited. I recommend the following one: “Application of NSGA-II algorithm and Kriging model to optimise the process parameters for improvement the quality of Fresnel lenses”. 

2. Line 77: “Coating on optical polymers…”

3. Lines 80-81: “…with a thin film which thickness is equal to a quarter of the wavelength.”

4. Line 122: English correction is required.

5. Line 272 may be deleted. 

6. Lines 281-283: Sense is incorrect and should be revisited.

7. Fig. 3: Is a copyright permission needed?

8. Variables should be written in Italics. Indexes for the variables should be corrected.

9. Line 501: “…too high pressure…”.

10. Lines 518-526 should be replaced.

Author Response

Please see enclosed, thank you.

This manuscript is a resubmission of an earlier submission. The following is a list of the peer review reports and author responses from that submission.

Round 1

Reviewer 1 Report

This paper studies the manufacturing parameters of Fresnel lens produced by the IMD in-mold decoration technology. The kriging model of process parameters and optimization objectives is established, leading to best values of process temperature and pressure.

To my opinion, this is not a relevant scientific paper and does not deserve publication. It is rather a “mixed salad” made of several, textbook-like pieces of trivial information, many qualitative and confused considerations, and a little bit of quantitative results:

-       the introduction is a long (for pages!), useless text where well-known information is presented;

-       the literature review section could well appear in a review paper. This, however, is a research paper!

-       the material section gives absolutely no information on the actual polymer used. We only learn that it is some PMMA and we read (uselessly!) all arguments why it should be preferred to other materials. Incidentally, on line 379 the authors claim that shear rate is measured in m/s!

-       same story for the methods section. You must explain the process you used, and not discuss academically which process is better. At the end of sections 3 and 4 the reader just knows nothing about the properties of the polymer used and the apparatus used to obtain the results. Incidental, but very important: where does Table 3 come from? there no indication on how data are collected, if they are experiments or simulation, and so on. Put in such a way, Table 3 is just a collection of numbers coming from nowhere;

-       the case study presentation in very confused: first, the real process apparatus is presented (incidentally, in the text it is said that Figure 5 shows the Fresnel lens, whereas Figure 5 is actually a picture of the injection molding machine); then, the comparison between different process techniques is again done; next the results of some simulations are presented (which software, what conditions, there is no information on how the simulations were made); next, some very generic consideration on the role of pressure and temperature are made. Finally some simulation results are presented and from them the optimization is implemented. On this last point, I notice the singularity of getting optimization conditions with two decimal digits (pressure 320.35MPa, temperature 251.40°C), while temperature and pressure in the simulation were changed by steps of the order of 10 MPa and 10°C, respectively.

-       in the results and discussion section, again very general considerations are made, which are not useful for the purpose of the work.

In the end, I am convinced that the paper cannot be accepted for consideration.

English is definitely below publication standards. There are many mistakes, misprints, grammar and syntax errors

Author Response

Please kind see enclosed, thank you.

Reviewer 2 Report

“Based on NSGA-II applied Kriging model relationship between

transmittance to lens with Fresnel to improve the influence of

structural node displacement “investigates and optimizes process parameter in manufacturing Fresnel lens via injection molding by comparing three different methods: double-stage injection molding, in Mold decoration and Injection compression molding.

Revision or comment

The work is interesting, well introduced, performed, and proposed. It seems that the paper is in a final reviewing stage and hence no relevant question have been intercepted.

Minor text editing of the language is useful to clear some sentence as follows:

line 439: “The third way we try to use the double stage injection molding method”, Is a verb missing?

Author Response

Please kind see enclosed, thank you.

Reviewer 3 Report

The work contains detailed results of modeling the injection molding process, which are certainly useful for Fresnel lens technology. In addition to modeling a real Fresnel lens was produced in the work, its transmission spectrum is presented. The work may be useful to specialists in the field of optical technology. However, the manuscript is completely unsuitable for publication in Polymers in the current version and should be carefully revised and rewritten. My comments are below.

1.     English requires significant editing with the participation of a specialist who is fluent in English. Then, the manuscript must be edited by a native English speaker or a special English Editing Service. The manuscript is very difficult to read and understand in the presented version. In particular, the Title should be rewritten in more correct words. Unfortunately, the fragments rewritten according to the recommendations of the first reviewer are also far from perfect.

2.     The proper style of scientific publication is almost never observed. In many cases, the authors should use synonyms rather than repeating the same word. Many sentences should be simplified. I don't see the point in highlighting specific grammatical and stylistic errors, since they occur throughout the manuscript.

3.     The structure of the manuscript should be improved. For example, lines 680-693 are, I suppose, misplaced and could be moved into the Introduction.

4.     Abbreviations are often used without descriptions, but some abbreviations are descripted many times.

5.     All variables should be written in Italics.

6.     Figure 7 is not needed in the revised manuscript.

7.     Transmission spectrum in Figure 14 is of very low quality. Probably, the detector change (for example, PMT was changed to PbS) led to a big noise because of small changes in transmission. It is not clear why the transmission reaches 95.4%, corresponding to a plane-parallel plate with n=1.36? What was the meaning of the transmission spectrum recording? Has the influence of the studied factors (node displacement and residual stress) on the transmittance been evaluated?

8. Have any other optical experiments been carried out to study real Fresnel lenses besides recording the transmission spectrum? As well, I suppose that in the manuscript it would be appropriate to compare the experimentally measured surface profile of the Fresnel lens with the required one.

Author Response

Please see enclosed, thank you.

Round 2

Reviewer 1 Report

The authors made profound changes to the manuscript, which I consider sufficient for its acceptance

English still needs some revision

Author Response

Please see enclosed, thank you.
